# Network Pharmacology-Based Characterization of Mecasin (KCHO-1) as a Multi-Target Modulator of Neuroinflammatory Pathways in Alzheimer’s Disease

**DOI:** 10.3390/nu18010008

**Published:** 2025-12-19

**Authors:** Hyein Jo, Joonyoung Shin, Hyorin Lee, Gi-Sang Bae, Sungchul Kim

**Affiliations:** 1Institute for Global Rare Disease Network, Professional Graduate School of Korean Medicine, Wonkwang University, Iksan 54538, Republic of Korea; fkquf0127@naver.com (H.J.); spm1219@naver.com (J.S.); dlgyfls319@naver.com (H.L.); 2Department of Pharmacology, School of Korean Medicine, Wonkwang University, Iksan 54538, Republic of Korea; baegs888@wku.ac.kr; 3Research Center of Traditional Korean Medicine, Wonkwang University, Iksan 54538, Republic of Korea

**Keywords:** Mecasin, Alzheimer’s disease, network pharmacology, KEGG, molecular docking

## Abstract

**Background/Objectives**: Mecasin (KCHO-1) is a standardized multi-herb formulation containing diverse bioactive compounds predicted to engage multiple molecular targets. This study applied an integrative network pharmacology approach to explore how Mecasin may interact with Alzheimer’s disease (AD)-related molecular networks. **Methods**: Bioactive constituents from 9 herbs were screened through OASIS and PubChem, and their predicted targets were cross-referenced with 8886 AD-associated genes from GeneCards. Overlapping genes were analyzed using protein–protein interaction mapping, Gene Ontology, and KEGG to identify potential Mecasin–AD core nodes and pathways. Co-expression, co-regulation, and molecular docking analyses were performed to further characterize mechanistic relevance. **Results**: Network integration identified 6 core genes—AKT1, STAT3, IL6, TNF, EGFR, and IL1B—positioned within signaling pathways related to neuronal survival, inflammatory regulation, and cellular stress responses, including FoxO, JAK–STAT, MAPK, and TNF pathways. Molecular docking suggested that several Mecasin compounds may interact with targets such as AKT1 and TNF. **Conclusions**: These in silico findings indicate that Mecasin, a multi-component formulation containing numerous phytochemicals that generate broad compound–target associations, may interface with interconnected neuroimmune pathways relevant to AD. While exploratory, the results highlight potential multi-target mechanisms that merit further investigation and provide a systems-level framework to inform future experimental validation.

## 1. Introduction

Dementia is a growing global health concern, with Alzheimer’s disease (AD) representing its most prevalent form and a major contributor to morbidity in aging populations [1,2,3,4]. AD pathology involves amyloid and tau abnormalities as well as interconnected processes such as neuroinflammation and oxidative stress, reflecting its multifactorial nature [5]. Current therapeutic options provide only symptomatic relief and have limited impact on disease progression [6]. These limitations highlight the need to elucidate the mechanisms underlying AD and to explore multi-pathway therapeutic strategies that can modulate its complex pathological networks [7].

Mecasin (KCHO-1) is a standardized herbal formulation composed of *Curcuma longa*, *Polygala tenuifolia*, *Gastrodia elata*, *Salvia miltiorrhiza*, *Paeonia lactiflora*, *Glycyrrhiza uralensis*, *Pseudocydonia sinensis*, *Aconitum carmichaeli*, and *Atractylodes japonica*. It has been designated as an orphan herbal investigational product for amyotrophic lateral sclerosis (ALS) and has shown clinical promise by improving functional outcomes and slowing disease progression in ALS patients [8]. Considering that ALS and AD are neurodegenerative disorders sharing multiple neuroinflammatory features [9], assessing Mecasin in an AD-related mechanistic context is relevant. Given its documented neuroprotective and anti-inflammatory properties, Mecasin may influence AD-associated neuroimmune pathways implicated in cognitive decline. On this basis, the present study employed an integrative network-based pharmacological framework to explore the multi-target mechanisms through which Mecasin may exert disease-modifying actions in AD.

Conventional drug-development strategies in AD typically focus on single molecular targets, despite the multifactorial nature of neurodegenerative pathology. This reductionist paradigm poses inherent limitations in addressing the interconnected cascades of oxidative stress, synaptic dysfunction, and neuronal loss. Network pharmacology has emerged as a systems-level analytical framework that integrates chemical, genomic, and pharmacological information to uncover pathway networks relevant to disease mechanisms [10]. By mapping drug-herb-component–target–pathway relationships within complex biological systems, this approach enables the identification of compounds capable of modulating multiple pathological axes simultaneously and is therefore particularly well-suited for elucidating the mechanistic basis of multi-component herbal formulations [11].

In this study, we adopted an in silico network pharmacology approach to explore how Mecasin may engage molecular pathways implicated in AD. By integrating compound profiles, predicted targets, and AD-related gene networks, we examined the possible mechanisms through which Mecasin could influence neuroinflammatory and neuroprotective processes. This exploratory analysis outlines potential pathways that Mecasin may engage within the complex biology of AD.

## 2. Materials and Methods

### 2.1. Collection of Active Compounds from Mecasin

To construct the Mecasin-associated molecular network, phytochemical constituents of each herbal component were retrieved from a publicly available traditional medicine database. The list of ingredients was acquired by querying the OASIS Traditional Medicine Information Portal (http://oasis.kiom.re.kr/, accessed on 5 January 2024) [12]. For each compound, physicochemical profiles and corresponding PubChem compound identifiers (CID) were extracted to enable downstream target-prediction and annotation [13].

### 2.2. Identification of Compound-Associated Target Genes

Active phytochemicals identified from the nine herbal components of Mecasin (*Curcuma longa*, *Polygala tenuifolia*, *Gastrodia elata*, *Salvia miltiorrhiza*, *Paeonia lactiflora*, *Glycyrrhiza uralensis*, *Pseudocydonia sinensis*, *Aconitum carmichaeli*, and *Atractylodes japonica*) were queried in PubChem to obtain putative protein-coding targets. For each compound, gene-target associations were retrieved from the PubChem “Chemical-Gene Co-Occurrences in Literature” dataset. Compounds lacking available literature-based target annotations were excluded. Duplicate and erroneous compound entries were removed through manual curation. The final number of compound-associated gene targets included 292 for *Curcuma longa*, 818 for *Polygala tenuifolia*, 322 for *Gastrodia elata*, 1050 for *Salvia miltiorrhiza*, 2555 for *Paeonia lactiflora*, 2719 for *Glycyrrhiza uralensis*, 1289 for *Pseudocydonia sinensis*, 329 for *Aconitum carmichaeli*, and 358 for *Atractylodes japonica*.

### 2.3. Retrieval of AD-Associated Genes and Intersection with Mecasin Targets

To investigate the relevance of Mecasin to AD pathophysiology, AD-associated genes were retrieved from the GeneCards database (https://www.genecards.org/, accessed on 5 January 2024) using the keyword “Alzheimer’s Disease” A total of 8886 AD-related genes were collected [14]. Subsequently, Mecasin-associated target genes were intersected with the AD gene set to identify shared targets.

### 2.4. Protein–Protein Interaction Network Construction and Core Gene Identification

Overlapping Mecasin–AD target genes were imported into the STRING database to construct a protein–protein interaction (PPI) network [15]. Interaction data with high confidence scores were retrieved and subsequently analyzed using Cytoscape (version 3.10.1; http://cytoscape.org/, accessed on 10 January 2024) [16]. To identify core genes, network topology metrics were computed, including degree centrality (DC), betweenness centrality (BC), and closeness centrality (CC) [17,18]. Genes with values above the mean threshold for each metric were retained during an initial screening step, and this threshold-based filtering procedure was repeated iteratively. Genes consistently enriched across successive screening rounds were defined as hub genes, representing key molecular nodes potentially mediating Mecasin’s therapeutic effects in AD.

### 2.5. Functional Enrichment and Network Construction

Functional enrichment analysis was performed to elucidate biological processes and signaling pathways associated with Mecasin-related targets. Gene Ontology (GO) biological process and Kyoto Encyclopedia of Genes and Genomes (KEGG) pathway enrichment were conducted using the Enrichr platform (http://maayanlab.cloud/Enrichr, accessed on 12 January 2024) [19,20,21]. Enriched terms were ranked according to statistical significance, and pathways meeting the predefined *p*-value threshold were retained for interpretation [22,23]. To visualize the multilevel interaction structure of Mecasin within the AD context, a drug–herb–compound–target–pathway (D-H-C-T-P) network was constructed [24]. Data obtained from compound screening, target identification, AD-related gene retrieval, and PPI–core gene analysis were integrated and imported into Cytoscape.

### 2.6. Co-Expression, Co-Regulation, and Molecular Docking Analysis of Core Targets

To further characterize the mechanistic relevance of the core genes, we performed co-expression, co-regulation, and molecular docking analyses. Co-expression analysis was performed in STRING to evaluate the combined scores among the 6 core genes. For proteomic co-regulation assessment, we utilized ProteomeHD [25]. Because TNF is a secreted ligand and not consistently represented in proteome-wide co-regulation datasets, TRAF2 was employed as a proxy to interrogate TNFR1/2–TRAF2 signaling, with results interpreted at the receptor–adaptor level rather than ligand abundance. For structural validation, molecular docking simulations were performed between Mecasin-derived bioactive compounds and proteins encoded by the core genes. Docking was carried out using CB-Dock2 (https://cadd.labshare.cn/cb-dock2/index.php, accessed on 1 September 2025), which enables blind-docking cavity prediction and ligand-binding evaluation [26]. Protein structures (PDB format) and compound files (SDF format) were obtained from the PDB (https://www.rcsb.org, accessed on 1 September 2025) and PubChem (https://pubchem.ncbi.nlm.nih.gov, accessed on 1 September 2025), respectively [27]. Prior to docking, the protein structures were preprocessed in CHIMERA (v1.19) by removing the original solvent and bound ligands, followed by the addition of hydrogens and charges to optimize the receptor conformation for ligand-binding analysis [28].

## 3. Results

### 3.1. Identification of Mecasin Bioactive Constituents and Predicted Targets

The workflow of the entire study is shown in Figure 1. We first curated the bioactive constituents of Mecasin using the OASIS database. A total of 198 phytochemical candidates were initially retrieved. Following the removal of compounds lacking PubChem identifiers, duplicates, and molecules without defined gene targets, 192 active constituents were retained (Table 1). These included 6 from *Curcuma longa*, 20 from *Polygala tenuifolia*, 5 from *Gastrodia elata*, 17 from *Salvia miltiorrhiza*, 34 from *Paeonia lactiflora*, 83 from *Glycyrrhiza uralensis*, 17 from *Pseudocydonia sinensis*, 5 from *Aconitum carmichaeli*, and 5 from *Atractylodes japonica*. Cross-referencing these bioactive molecules with target-gene databases yielded 1913 predicted Mecasin-associated targets (Figure 2A).

### 3.2. Intersection Between Mecasin-Associated Targets and Alzheimer’s Disease Gene Networks

To evaluate the therapeutic relevance of Mecasin in the context of AD, we compared Mecasin-associated gene targets with AD-related genetic signatures curated from the GeneCards database (8886 AD-associated genes). Of the 1913 predicted Mecasin target genes, 942 overlapped with AD-related genes (Figure 2B, Appendix A).

### 3.3. Analysis of Key Genes and Networks Associated with Mecasin and Alzheimer’s Disease

Based on the 942 overlapping Mecasin—AD targets, we reconstructed a PPI network comprising 942 nodes and 9948 edges (Figure 3). Network topology analysis was then conducted in Cytoscape (v3.10.1) using degree centrality (DC), betweenness centrality (BC), and closeness centrality (CC) as selection metrics. In the first screening step, thresholds of DC > 21.12102, BC > 2.09 × 10^−3^, and CC > 0.336771 were applied, yielding a refined network of 174 nodes and 2910 edges. A second iteration using DC > 33.44828, BC > 0.005282, and CC > 0.531056 further narrowed candidate nodes (Figure 3A,B), followed by a third filtering round (DC > 27.75, BC > 0.007591, CC > 0.785929625, Figure 3C). The final screening step (DC > 33.42857, BC > 0.012156, CC > 0.878397) resulted in 6 core genes—AKT1, STAT3, IL6, TNF, EGFR, and IL1B—forming a condensed interaction network of 6 nodes and 15 edges (Figure 3D and Table 2). These hubs represent key regulators of inflammatory signaling, apoptosis, neuronal survival, and metabolic processes implicated in AD progression.

### 3.4. Functional and Pathway Enrichment Analysis of Mecasin–AD Core Genes

To investigate the functional mechanisms through which Mecasin may act in AD, the 6 core genes identified from our network analysis were submitted to Enrichr for GO enrichment analysis and KEGG pathway annotation (Figure 4). The GO enrichment results revealed 589 terms in the Biological Process (BP) category, 25 terms in the Cellular Component (CC) category, and 47 terms in the Molecular Function (MF) category. Based on *p*-values, the top 10 terms from each category were further examined. In the BP category, key enriched terms included “Negative regulation of catabolic process” and “Positive regulation of miRNA transcription.” In the CC category, enriched terms were related to structures such as “Multivesicular body, internal vesicle” and “Intracellular vesicle,” while the MF category included terms such as “Cytokine activity” and “Receptor ligand activity.” KEGG pathway identified 144 associated signaling pathways. Among the top 30 pathways ranked by *p*-value, eight were closely related to AD mechanisms: the Toll-like receptor signaling pathway, C-type lectin receptor signaling pathway, HIF-1 signaling pathway, TNF signaling pathway, FoxO signaling pathway, JAK–STAT signaling pathway, MAPK signaling pathway, and adipocytokine signaling pathway (Figure 5).

### 3.5. D-H-C-T-P Network Analysis of Mechanisms of Mecasin in AD

To integrate pharmacological components with disease-relevant molecular pathways, a D-H-C-T-P network was constructed (Figure 6). This multilevel network revealed that Mecasin’s therapeutic activity is convergently directed toward key AD-related core genes, including TNF, IL6, and AKT1. Mecasin’s major bioactive constituents showed broad connectivity to AD-related core genes. Curcumin was associated with 5 core genes, glycyrrhetic acid with 5, Tanshinone IIA with 4, tenuifolin with 4, albiflorin with 4, paeoniflorin with 4, salvianolic acid B with 3, and liquiritigenin with 3. TNF, IL6, and AKT1 emerged as shared targets across all 7 compounds. AKT1 was the only gene associated with all 8 Mecasin–AD core KEGG pathways. IL6 was associated with 6 pathways, TNF with 5, EGFR with 4, IL1B with 4, and STAT3 with 3.

### 3.6. Co-Expression, Co-Regulation, and Molecular Docking Analysis of Core Targets

To further validate the mechanistic relevance of Mecasin–AD core genes, we performed integrated analyses combining gene co-expression, co-regulation, and molecular docking (Figure 7). Among the 6 core Mecasin–AD genes, the STRING protein–protein interaction network showed a high level of functional connectivity, with an average combined score of 0.933 across all pairwise interactions. Notably, the interaction strengths between EGFR–STAT3 (0.998), IL1B–TNF (0.998), and IL1B–IL6 (0.996) were among the highest, reflecting tightly linked regulatory relationships within inflammatory and signaling pathways (Figure 7A). Co-expression analysis of the 6 core Mecasin–AD genes revealed coordinated transcriptional patterns among key inflammatory mediators. The co-expression scores were 0.261 for TNF–IL6, 0.515 for IL1B–IL6, and 0.616 for TNF–IL1B. Notably, TNF demonstrated the highest centrality within this co-expression module (Figure 7B). Using a cut-off threshold of 0.945, co-regulation analysis identified genes co-regulated with TRAF2—a key adaptor in TNF signaling—including INTS12, PLEKHA2, CNOT9, ZNF143, and MITD1 (Figure 7C). The average percentile score of these co-regulated genes was 0.963, with an average co-regulation score of 0.171, reflecting tight regulatory integration within the TNF signaling network. Molecular docking was conducted for 6 protein–ligand pairs consisting of two core proteins—AKT1, associated with neuroplasticity, and TNF, associated with inflammatory signaling—and 3 Mecasin compounds (curcumin, salvianolic acid B, and tanshinone IIA) (Figure 7D–I). For AKT1, the binding affinities were −9.5 kcal/mol with curcumin, −8.4 kcal/mol with salvianolic acid B, and −11.7 kcal/mol with tanshinone IIA. For TNF, the binding affinities were −7.7 kcal/mol with curcumin, −8.4 kcal/mol with salvianolic acid B, and −9.0 kcal/mol with tanshinone IIA.

## 4. Discussion

AD is the most prevalent form of dementia and represents a progressive neurodegenerative disorder characterized by the gradual loss of neuronal structure and function [29,30]. Clinically, AD manifests as a continuum from subtle short-term memory deficits and spatial disorientation in the early stages to pronounced cognitive impairment, behavioral disturbances, and ultimately complete dependence in daily activities during the late phase. The AD course is marked by progressive synaptic dysfunction, neuronal death, and widespread network disintegration, leading to a profound decline in memory, reasoning, and executive function [31]. Although the exact pathological mechanisms underlying AD remain unclear, abnormal overproduction and aggregation of Aβ peptides are believed to play a critical role in disease progression. The resulting extracellular amyloid plaques interfere with neuronal communication, induce neuroinflammatory responses, and promote oxidative and metabolic stress. These events impair synaptic function and contribute to neuronal degeneration, ultimately driving the cognitive and behavioral manifestations of AD [32,33].

Mecasin was developed as an orphan drug candidate for the treatment of ALS, with the aim of evaluating its safety and efficacy in alleviating neurodegenerative symptoms. Preclinical investigations have demonstrated that Mecasin, a standardized herbal formulation, exerts both neuroprotective and anti-neuroinflammatory effects. Its safety has been consistently confirmed across in vitro and in vivo experimental models. In lipopolysaccharide-stimulated BV2 microglia, Mecasin markedly attenuated the expression of inflammatory mediators (iNOS, COX-2), and pro-inflammatory cytokines (TNF-α, IL-1β, IL-6) through suppression of NF-κB nuclear translocation and IκB-α phosphorylation. Mechanistically, it upregulated heme oxygenase-1 (HO-1) expression via Nrf2 activation, enhancing endogenous antioxidant defense. Comparative analyses further revealed that key constituents such as Curcuma longa and Polygala tenuifolia synergistically contributed to this anti-neuroinflammatory action [34,35]. In HT22 neuronal cells, Mecasin protected against glutamate- and hydrogen peroxide-induced cytotoxicity in a concentration-dependent manner. This protection was mediated through ERK/p38-MAPK–dependent activation of Nrf2 and subsequent HO-1 induction, leading to suppression of intracellular ROS and maintenance of neuronal viability [36]. Toxicological assessments confirmed Mecasin’s favorable safety profile. Single-dose intravenous and intramuscular pharmacopuncture studies in Sprague–Dawley rats showed no mortality, clinical toxicity, or histopathological abnormalities up to 2000 mg/kg [37,38]. Furthermore, chronic 26-week oral administration produced no adverse clinical, hematological, or organ-specific findings, establishing a NOAEL exceeding 2000 mg/kg/day [39]. In the hSOD1G93A transgenic mouse model of ALS, Mecasin delayed disease onset, improved motor coordination, and prolonged survival. Histological analysis revealed reduced spinal motor neuron loss, suppressed microglial activation, and decreased oxidative markers such as iNOS and gp91phox. These effects were linked to downregulation of ERK1/2 phosphorylation and inhibition of the NADPH oxidase–MAPK axis [40]. Collectively, these findings establish Mecasin as a safe and mechanistically robust herbal formulation with dual actions on redox and inflammatory homeostasis. Through modulation of Nrf2/HO-1, NF-κB, and MAPK signaling, Mecasin mitigates glial activation, oxidative stress, and neuronal degeneration. These preclinical insights provide a mechanistic rationale for its therapeutic potential in neurodegenerative disorders, including ALS and AD. Clinically, a recent multicenter, double-blind, placebo-controlled phase IIa trial further supported these findings, showing that 12 weeks of Mecasin treatment slowed functional decline in ALS patients without serious adverse events [8]. Building upon these preclinical and clinical observations, we hypothesized that Mecasin’s neuroprotective and anti-inflammatory mechanisms may also extend to AD. Therefore, this study employed a network pharmacology approach to explore the diverse ingredient–target interactions underlying Mecasin’s potential regulatory effects on Alzheimer’s pathology.

Using the OASIS database, a compound–target network was constructed based on the active ingredients of Mecasin, comprising 942 nodes and 9948 edges. To assess the disease relevance of these targets, 8886 AD–related genes retrieved from GeneCards were cross-analyzed with 1913 Mecasin-associated targets. The intersection revealed 942 genes that overlapped with AD-associated gene sets. Topological network analysis (degree centrality, betweenness centrality, and closeness centrality) was applied in three successive screening rounds to identify hub genes. 6 key targets were confirmed: STAT3, IL1B, TNF, IL6, EGFR, and AKT1. Functional enrichment analysis was then performed to elucidate the biological implications of these hub genes. In the GO Biological Process category, enriched terms included positive regulation of gene expression and negative regulation of apoptotic process. KEGG pathway analysis revealed major signaling cascades potentially involved in Mecasin’s AD-modulating effects: the HIF-1 signaling pathway, JAK–STAT signaling pathway, TNF signaling pathway, and MAPK signaling pathway. Together, these findings complement the existing evidence for Mecasin and suggest a consistent mechanistic outline through which its multi-ingredient formulation may relate to AD pathology. On this basis, three possible mechanistic aspects through which Mecasin may influence AD can be proposed: neuroprotection, neuroregenerative support, and anti-inflammatory regulation.

Neuronal loss and impaired survival signaling are central pathological features of AD, with accumulating evidence indicating that disruptions in PI3K–AKT, JAK–STAT, and MAPK pathways contribute significantly to synaptic degeneration, oxidative injury, and apoptosis [41,42,43,44]. In our network pharmacology analysis, Mecasin was found to converge on several key nodes associated with neuroprotective processes, particularly AKT1, STAT3, and EGFR, suggesting that the formulation may engage molecular modules that support neuronal resilience in AD. Among the 6 Mecasin–AD core genes identified, AKT1 emerged as a major regulatory hub, aligning with prior studies showing that AKT-mediated phosphorylation cascades protect neurons from Aβ-induced toxicity, mitochondrial dysfunction, and excitotoxic damage [45,46,47]. The involvement of STAT3 and EGFR further reinforces this neuroprotective framework, as both genes have been implicated in promoting neuronal survival, dendritic maintenance, and glial-mediated trophic support [48,49,50]. KEGG pathway analysis highlighted JAK–STAT, HIF-1, and MAPK signaling as major functional clusters, all of which have well-established roles in protecting neurons under oxidative or inflammatory stress [51,52,53]. These pathways coordinate anti-apoptotic signaling, enhance mitochondrial stability, and regulate antioxidant gene expression, suggesting that Mecasin may potentially engage these protective mechanisms. The molecular docking results provided preliminary indications of a potential neuroprotective role of Mecasin. Tanshinone IIA exhibited strong affinity toward AKT1, followed by curcumin and salvianolic acid B, consistent with previous reports demonstrating that these phytochemicals activate AKT-related regenerative and protective pathways [54]. Curcumin and salvianolic acid B, in particular, have been shown to modulate oxidative pathways, inhibit ROS formation, and stabilize synaptic proteins, which may complement Mecasin’s overall neuroprotective profile [55,56,57]. Although these in silico findings do not confirm functional activity, they outline plausible molecular interfaces through which Mecasin compounds could engage neuroprotective signaling in AD. Collectively, these multi-level observations suggest that Mecasin may exert neuroprotective effects by modulating AKT1-centered survival pathways and by interacting with STAT3- and EGFR-dependent trophic mechanisms. These proposed actions align with previously reported anti-apoptotic, antioxidant, and mitochondrial-stabilizing effects of Mecasin’s constituent herbs and compounds. Further experimental validation is warranted to determine whether these network-level predictions translate into functional protection against neuronal injury in vivo.

Beyond neuronal survival, accumulating evidence indicates that impairment of adult neurogenesis and synaptic remodeling is a critical component of cognitive decline in AD. Reduced hippocampal neurogenesis and dysregulated neurotrophic signaling, including BDNF/TrkB and PI3K–AKT–CREB pathways, have been linked to memory impairment and increased vulnerability of neuronal circuits in AD and related conditions [58]. In our KEGG pathway analysis, several Mecasin–AD core genes (IL6, AKT1, STAT3) appeared within signaling pathways associated with neuronal remodeling. Among these, the FoxO and MAPK pathways are known to participate in axon extension, dendritic growth, and activity-dependent synaptic plasticity [59,60,61,62]. Taken together, these associations suggest that Mecasin may have relevance to regenerative processes beyond basic survival support, raising the possibility of a connection to structural adaptations within vulnerable neuronal circuits.

Consistent with the inflammatory burden characteristic of AD, Mecasin–AD core genes showed strong representation across TNF signaling, Toll-like receptor signaling, and C-type lectin receptor pathways, with IL6, IL1B, TNF, and AKT1 appearing across all three. These convergent pathways underscore the central role of cytokine-driven cascades in shaping microglial activation and neuroinflammatory responses [63,64]. While our co-regulation analysis identified TRAF2 as a point of convergence within these networks—together with a small cluster of genes displaying transcriptional coherence—this finding is best interpreted as a preliminary indication of how receptor-proximal TNF signaling may interface with downstream regulatory processes. Such patterns, together with the favorable docking affinities of Mecasin compounds for TNF, suggest that Mecasin may modulate inflammatory signaling at multiple levels, potentially contributing to the attenuation of cytokine-driven neuroinflammation in AD.

This study has several methodological limitations inherent to the in silico network pharmacology approach. First, although compound–target prediction tools and basic pharmacokinetic filters were applied, these methods cannot fully reflect the absorption, distribution, metabolism, and excretion characteristics of Mecasin’s diverse phytochemicals, nor do they ensure biologically relevant target engagement. Second, because Mecasin is a multi-herb formulation containing numerous phytochemicals, it naturally generates a large number of predicted compound–target associations, and its pharmacological effects are likely to arise from synergistic interactions among components. While network inference can identify convergent mechanistic hubs within this complexity, it cannot establish causal contributions of individual herbs or compounds to AD-related pathways. Third, PPI-based core-gene extraction and pathway mapping rely on heterogeneous database evidence, meaning that the prominence of nodes such as AKT1, IL6, TNF, STAT3, EGFR, and IL1B may partially reflect their general network centrality rather than Mecasin-specific actions. Fourth, the molecular docking analysis represents an exploratory structural assessment and is subject to notable constraints, as it was performed as single docking runs on CB-Dock2 without comparison to known ligands, without incorporating receptor flexibility, and without consideration of ADME/BBB properties of the compounds. Docking scores alone do not establish physical binding or biological efficacy, particularly for multi-component herbal mixtures, and the results should therefore be interpreted solely as plausible structural poses rather than evidence of functional activity. Finally, the mechanistic predictions generated from this network analysis will require targeted molecular or in vivo studies to confirm the proposed interactions and pathways. Nevertheless, in silico network pharmacology analysis offers a useful framework for examining how complex herbal mixtures may relate to diverse aspects of disease biology. By identifying plausible Mecasin–AD interaction hubs and pathway clusters, the present study offers theoretical groundwork and directional guidance for future experimental investigations into the neuroimmune mechanisms of Mecasin.

## 5. Conclusions

This network pharmacology study explored potential connections between Mecasin and molecular features associated with AD. 6 Mecasin–AD core genes (AKT1, STAT3, IL6, TNF, EGFR, IL1B) emerged as central nodes linked to pathways involved in neuronal survival, regenerative signaling, and inflammatory regulation. KEGG enrichment highlighted FoxO, JAK–STAT, MAPK, and TNF signaling as potential axes of Mecasin activity. Docking results suggested that several Mecasin compounds may interact with AKT1 or TNF, although the relevance of these interactions to AD will require experimental verification. Overall, these in silico findings suggest possible multi-target interactions through which Mecasin may modulate interconnected neuroimmune mechanisms in AD, offering a systems-level framework that can inform subsequent mechanistic and in vivo validation.

## Figures and Tables

**Figure 1 nutrients-18-00008-f001:**
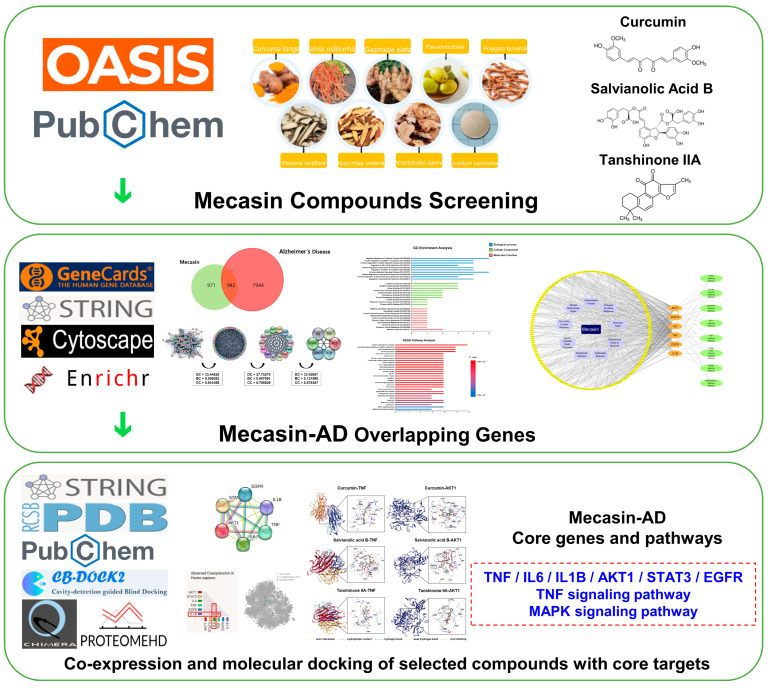
Workflow of the network pharmacology analysis of Mecasin in Alzheimer’s disease (AD). Bioactive compounds from the nine constituent herbs were retrieved from the OASIS and PubChem databases and mapped to predicted molecular targets. Mecasin-associated targets were then integrated with AD-related genes obtained from GeneCards to identify shared gene signatures. These overlapping genes were analyzed through STRING-based PPI network construction, followed by GO and KEGG enrichment analyses in Cytoscape and Enrichr to determine key functional nodes and pathways. Subsequent co-expression, co-regulation, and molecular docking assessments were conducted to examine the binding compatibility between core targets (TNF, AKT1) and Mecasin-derived compounds (curcumin, salvianolic acid B, Tanshinone IIA), outlining potential interactions relevant to AD-associated molecular mechanisms.

**Figure 2 nutrients-18-00008-f002:**
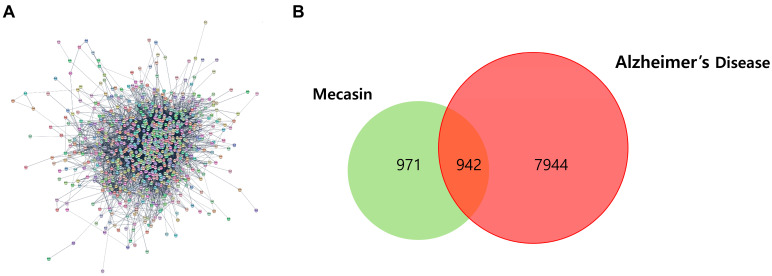
Network and overlapping target genes between Mecasin and Alzheimer’s disease (AD). (**A**) Protein–protein interaction (PPI) network constructed from Mecasin-associated target genes, comprising 942 nodes and 9948 edges. (**B**) Venn diagram showing the overlap between Mecasin-related targets and AD-related genes, identifying 942 shared targets.

**Figure 3 nutrients-18-00008-f003:**
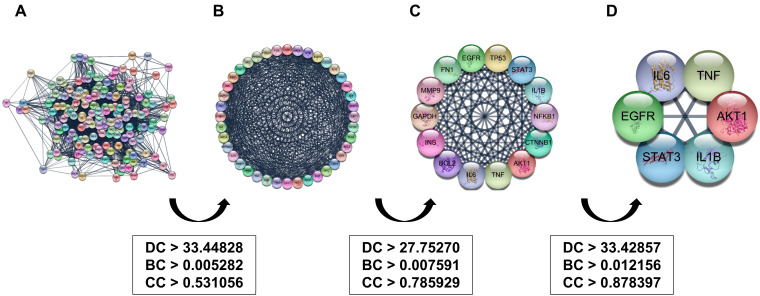
Topological refinement process for identifying core Mecasin–Alzheimer’s disease (AD) targets. (**A**) Protein–protein interaction (PPI) network constructed from the 942 overlapping Mecasin–AD genes. (**B**) First and second rounds of topological filtering based on degree, betweenness, and closeness centrality thresholds, reducing the network to intermediate candidate nodes. (**C**) Third screening step further narrowing high-ranking nodes according to increasingly stringent centrality criteria. (**D**) Final condensed network consisting of 6 core genes (AKT1, STAT3, IL6, TNF, EGFR, IL1B), each showing the highest topological importance across all filtering iterations.

**Figure 4 nutrients-18-00008-f004:**
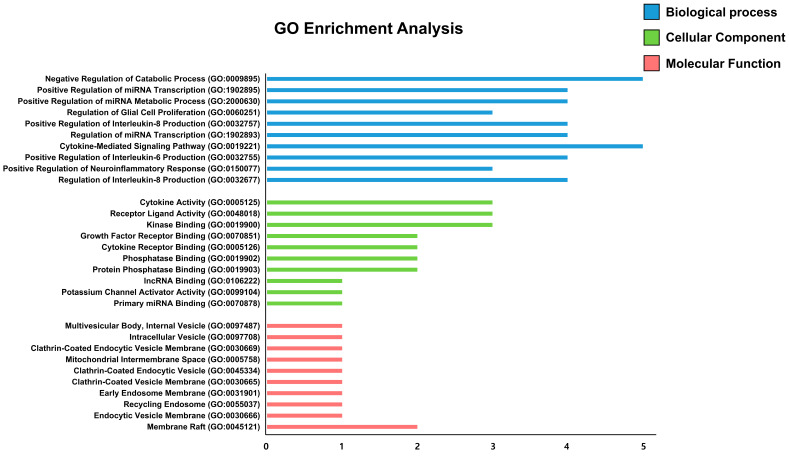
GO enrichment analysis of Mecasin-associated targets. Blue bars indicate Biological Processes (BP), green bars represent Cellular Components (CC), and red bars denote Molecular Functions (MF). The top-ranked GO terms in each category illustrate Mecasin’s potential involvement in gene expression regulation, intracellular organization, and protein-binding activity.

**Figure 5 nutrients-18-00008-f005:**
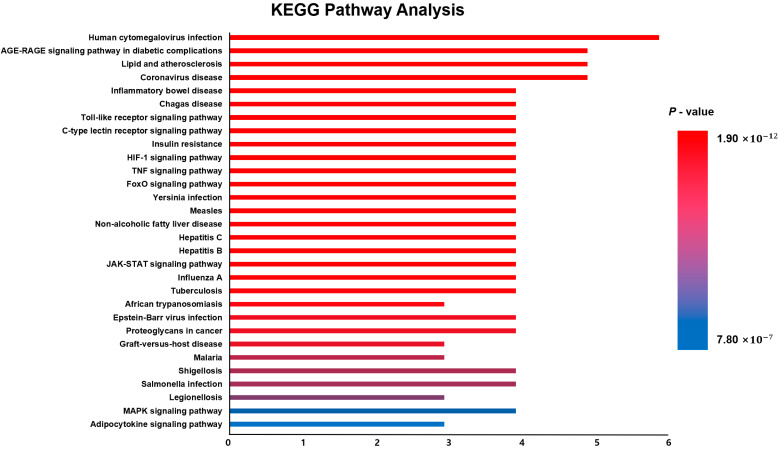
KEGG pathway analysis of Mecasin-associated targets. The core pathways include the HIF-1, JAK–STAT, TNF, and MAPK signaling pathways, indicating Mecasin’s potential involvement in regulating inflammatory, oxidative stress, and cell survival mechanisms related to Alzheimer’s disease.

**Figure 6 nutrients-18-00008-f006:**
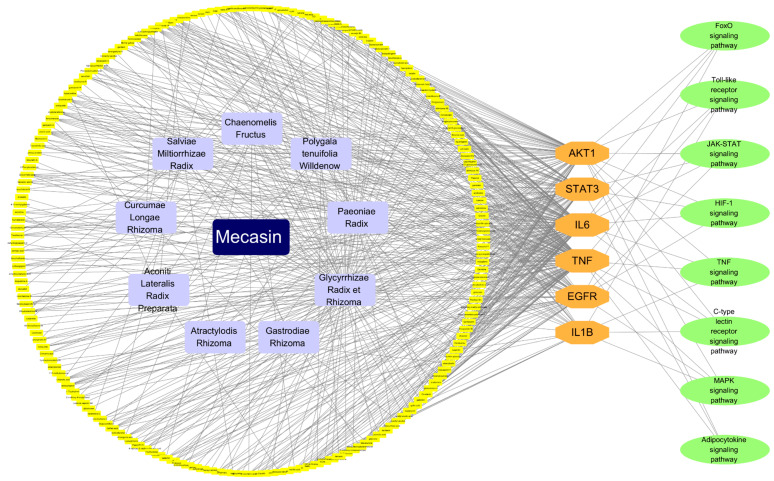
Drug–Herb–Compound–Target–Pathway (D-H-C-T-P) network of Mecasin in Alzheimer’s disease (AD). Navy nodes represent the drug (Mecasin), purple nodes indicate herbs, yellow nodes denote bioactive compounds, orange nodes correspond to target genes, and green nodes represent pathways. The network illustrates the multi-component and multi-target interactions through which Mecasin may exert regulatory effects on AD-related molecular mechanisms.

**Figure 7 nutrients-18-00008-f007:**
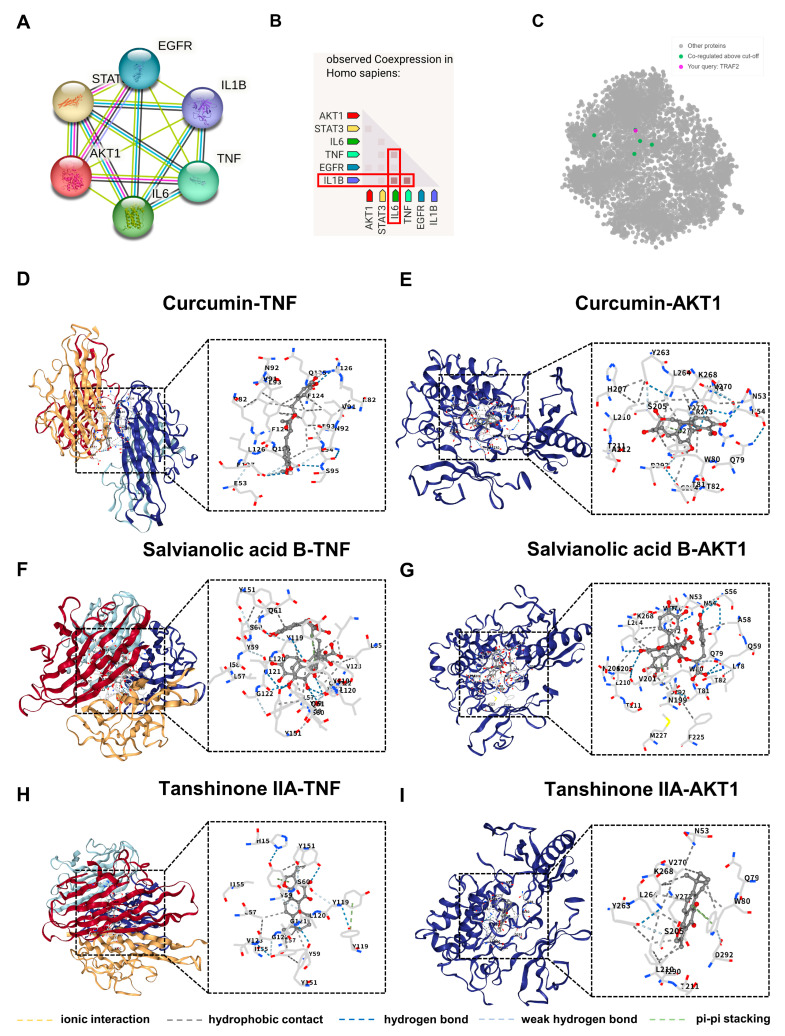
Co-expression, co-regulation, and molecular docking analyses of Mecasin–Alzheimer’s disease core targets. (**A**) Protein–protein interaction (PPI) network of the 6 core genes (AKT1, STAT3, IL6, TNF, EGFR, IL1B) showing high-confidence associations. (**B**) Co-expression profiling in Homo sapiens demonstrating coordinated transcriptional patterns among the 6 genes. (**C**) Co-regulation network centered on TRAF2, highlighting tightly associated regulatory partners within the TNF signaling axis. (**D**) Molecular docking of curcumin with TNF and (**E**) with AKT1. (**F**) Docking configuration of salvianolic acid B with TNF and (**G**) with AKT1. (**H**) Docking of tanshinone IIA with TNF and (**I**) with AKT1. Dashed-line colors indicate interaction types: ionic interactions (yellow), hydrophobic contacts (gray), hydrogen bonds (blue), weak hydrogen bonds (light blue), and pi–pi stacking (green).

**Table 1 nutrients-18-00008-t001:** List of bioactive compounds identified from Mecasin and their corresponding PubChem IDs.

Compound	Pubchem ID	Origin
Ar-turmerone	160512	*Curcumae Longa*
bisacurone	14287397	*Curcumae Longa*
curcumin	969516	*Curcumae Longa*
curzerene	12305301	*Curcumae Longa*
α-curcumene	442360	*Curcumae Longa*
α-turmerone	14632996	*Curcumae Longa*
3,4-dimethoxycinnamic acid	717531	*Polygala tenuifolia*
glomeratose A	11972358	*Polygala tenuifolia*
lancerin	5281645	*Polygala tenuifolia*
N-acetyl-D-glucosamine	439174	*Polygala tenuifolia*
senegin III	21669942	*Polygala tenuifolia*
sibiricose A5	102004867	*Polygala tenuifolia*
sibiricose A6	6326021	*Polygala tenuifolia*
tenuifoliside A	46933844	*Polygala tenuifolia*
tenuifoliside B	10055215	*Polygala tenuifolia*
tenuifoliside C	11968391	*Polygala tenuifolia*
1-(3,4-dimethoxyphenyl) ethan-1-one	14328	*Polygala tenuifolia*
2-hydroxybenzoic acid	338	*Polygala tenuifolia*
3,4,5-trimethoxycinnamic acid	735755	*Polygala tenuifolia*
3,6′-di-O-sinapoyl sucrose	11968389	*Polygala tenuifolia*
gentisin	5281636	*Polygala tenuifolia*
mangiferin	5281647	*Polygala tenuifolia*
onjisaponin F	10701737	*Polygala tenuifolia*
propyl benzoate	16846	*Polygala tenuifolia*
sucrose	5988	*Polygala tenuifolia*
tenuifolin	21588226	*Polygala tenuifolia*
vanillin	1183	*Gastrodia elata*
4-hydroxy-3-methoxybenzoic acid	8468	*Gastrodia elata*
benzyl alcohol	244	*Gastrodia elata*
hydroxybenzaldehyde	126	*Gastrodia elata*
vanillyl alcohol	62348	*Gastrodia elata*
Protocatechualdehyde	8768	*Salvia miltiorrhiza*
Tanshinone I	114917	*Salvia miltiorrhiza*
tanshindiol C	126072	*Salvia miltiorrhiza*
Miltirone	160142	*Salvia miltiorrhiza*
cryptotanshinone	160254	*Salvia miltiorrhiza*
Tanshinone IIA	164676	*Salvia miltiorrhiza*
danshenol A	3083514	*Salvia miltiorrhiza*
Rosmarinic acid	5281792	*Salvia miltiorrhiza*
Salvianolic Acid B	6451084	*Salvia miltiorrhiza*
salviolone	10355691	*Salvia miltiorrhiza*
arucadiol	11011966	*Salvia miltiorrhiza*
Danshensu	11600642	*Salvia miltiorrhiza*
deoxyneocryptotanshinone	15690458	*Salvia miltiorrhiza*
Dihydrotanshinone	5316743	*Salvia miltiorrhiza*
sugiol	94162	*Salvia miltiorrhiza*
Caffeic acid	689043	*Salvia miltiorrhiza*
Salvianic acid A	5281793	*Salvia miltiorrhiza*
Benzoic acid	243	*Paeonia lactiflora*
Carnitine	288	*Paeonia lactiflora*
Coumarin	323	*Paeonia lactiflora*
Gallic acid	370	*Paeonia lactiflora*
Glycyrrhizin	3495	*Paeonia lactiflora*
Protoporphyrin IX	4971	*Paeonia lactiflora*
L-Tryptophan	6305	*Paeonia lactiflora*
L(D)-Agrginin	6322	*Paeonia lactiflora*
Taurocholic acid	6675	*Paeonia lactiflora*
Methyl gallate	7428	*Paeonia lactiflora*
2-Phenylacetamide	7680	*Paeonia lactiflora*
Catechin	9064	*Paeonia lactiflora*
Glycocholic acid	10140	*Paeonia lactiflora*
Paeonol	11092	*Paeonia lactiflora*
Glycochenodeoxycholic acid	12544	*Paeonia lactiflora*
Glycyrrhizic acid	14982	*Paeonia lactiflora*
PGG	65238	*Paeonia lactiflora*
Catechin hydrate	107957	*Paeonia lactiflora*
Liquiritigenin	114829	*Paeonia lactiflora*
Paeoniflorin	425990	*Paeonia lactiflora*
Ononin	442813	*Paeonia lactiflora*
Cinnamic acid	444539	*Paeonia lactiflora*
Liquiritin	503737	*Paeonia lactiflora*
Cinnamaldehyde	637511	*Paeonia lactiflora*
2-methoxy cinnamaldehyde	641298	*Paeonia lactiflora*
Cinnamyl alcohol	5315892	*Paeonia lactiflora*
Isoliquiritin	5318591	*Paeonia lactiflora*
L-Palmitoylcarnitine	11953816	*Paeonia lactiflora*
6′-O-actylpaeoniflorin	21575212	*Paeonia lactiflora*
Mudanpioside C	21631098	*Paeonia lactiflora*
Oxypaeoniflorin	21631105	*Paeonia lactiflora*
Benzoylpaeoniflorin	21631106	*Paeonia lactiflora*
Albiflorin	24868421	*Paeonia lactiflora*
Galloyloxypaeoniflorin	71455849	*Paeonia lactiflora*
Glycyrrhetic acid	3230	*Glycyrrhiza uralensis*
18β-Glycyrrhetinic acid	10114	*Glycyrrhiza uralensis*
Glycyrrhizin	14982	*Glycyrrhiza uralensis*
Daidzin	107971	*Glycyrrhiza uralensis*
liquiritigenin	114829	*Glycyrrhiza uralensis*
licopyranocoumarin	122851	*Glycyrrhiza uralensis*
glabranin	124049	*Glycyrrhiza uralensis*
isoglycyrol	124050	*Glycyrrhiza uralensis*
galbridin	124052	*Glycyrrhiza uralensis*
uralsaponin B	163744	*Glycyrrhiza uralensis*
licoisoflavanone	392443	*Glycyrrhiza uralensis*
7-O-methyllutenone	441251	*Glycyrrhiza uralensis*
Schaftoside	442658	*Glycyrrhiza uralensis*
Vicenin-2	442664	*Glycyrrhiza uralensis*
Ononin	442813	*Glycyrrhiza uralensis*
dehydroglyasperin C	480775	*Glycyrrhiza uralensis*
gancaonin I	480777	*Glycyrrhiza uralensis*
6,8-Diprenylgenistein	480783	*Glycyrrhiza uralensis*
glycyrin	480787	*Glycyrrhiza uralensis*
Glyasperin C	480859	*Glycyrrhiza uralensis*
Liquiritin	503737	*Glycyrrhiza uralensis*
isoliquirigenin	638278	*Glycyrrhiza uralensis*
Isoschaftoside	3084995	*Glycyrrhiza uralensis*
Biochanin A	5280373	*Glycyrrhiza uralensis*
Formononetin	5280378	*Glycyrrhiza uralensis*
Kaempferol 3-O-methyl ether	5280862	*Glycyrrhiza uralensis*
kaempferol	5280863	*Glycyrrhiza uralensis*
genistein	5280961	*Glycyrrhiza uralensis*
Genkwanin	5281617	*Glycyrrhiza uralensis*
daidzein	5281708	*Glycyrrhiza uralensis*
Licoisoflavone A	5281789	*Glycyrrhiza uralensis*
luteone	5281797	*Glycyrrhiza uralensis*
pratensein	5281803	*Glycyrrhiza uralensis*
wighteone	5281814	*Glycyrrhiza uralensis*
Uralenol	5315126	*Glycyrrhiza uralensis*
Lupiwighteone	5317480	*Glycyrrhiza uralensis*
glabrone	5317652	*Glycyrrhiza uralensis*
Glycocoumarin	5317756	*Glycyrrhiza uralensis*
glycyrrhisoflavone	5317764	*Glycyrrhiza uralensis*
Isolicoflavonol	5318585	*Glycyrrhiza uralensis*
isoliquiritin	5318591	*Glycyrrhiza uralensis*
Kumatakenin	5318869	*Glycyrrhiza uralensis*
Licochalcone A	5318998	*Glycyrrhiza uralensis*
Licochalcone B	5318999	*Glycyrrhiza uralensis*
Licoricone	5319013	*Glycyrrhiza uralensis*
4′-O-methylgalbridin	5319664	*Glycyrrhiza uralensis*
glycyrol	5320083	*Glycyrrhiza uralensis*
licoisoflavone B	5481234	*Glycyrrhiza uralensis*
semilicoisoflavone B	5481948	*Glycyrrhiza uralensis*
gancaonin H	5481949	*Glycyrrhiza uralensis*
Licoflavonol	5481964	*Glycyrrhiza uralensis*
2′,4′,2-Trihydroxychalcone	5811533	*Glycyrrhiza uralensis*
homobutein	6438092	*Glycyrrhiza uralensis*
isoliquiritin apioside	6442433	*Glycyrrhiza uralensis*
echinatin	6442675	*Glycyrrhiza uralensis*
Licochalcone C	9840805	*Glycyrrhiza uralensis*
Kanzonol Y	10001604	*Glycyrrhiza uralensis*
liquiritin apioside	10076238	*Glycyrrhiza uralensis*
Licoarylcoumarin	10090416	*Glycyrrhiza uralensis*
2′,4′,2,4-Tetrahydroxychalcone	10107266	*Glycyrrhiza uralensis*
dehydroglyasperin D	10109594	*Glycyrrhiza uralensis*
glicoricone	10361658	*Glycyrrhiza uralensis*
Allolicoisoflavone B	10383349	*Glycyrrhiza uralensis*
Licochalcone D	10473311	*Glycyrrhiza uralensis*
licoleafol	11111496	*Glycyrrhiza uralensis*
Licoflavone B	11349817	*Glycyrrhiza uralensis*
Glabrol	11596309	*Glycyrrhiza uralensis*
11-Deoxoglycyrrhetinic acid	12305517	*Glycyrrhiza uralensis*
Licorice saponin A3	14187172	*Glycyrrhiza uralensis*
isoglycycoumarin	14187587	*Glycyrrhiza uralensis*
licoflavanone	14218028	*Glycyrrhiza uralensis*
2ʹ-hydroxyisolupalbigenin	14237659	*Glycyrrhiza uralensis*
isoderrone	14237660	*Glycyrrhiza uralensis*
licuraside	14282455	*Glycyrrhiza uralensis*
gancaonin L	14604077	*Glycyrrhiza uralensis*
Licorice saponin G2	14891565	*Glycyrrhiza uralensis*
Angustone A	15664151	*Glycyrrhiza uralensis*
Glyurallin A	15818598	*Glycyrrhiza uralensis*
isoangustone A	21591148	*Glycyrrhiza uralensis*
neoisoliquiritin	22524410	*Glycyrrhiza uralensis*
isolupalbigenin	26238934	*Glycyrrhiza uralensis*
Liquiritigenin 7,4′-di-O-glucopyranoside	46869260	*Glycyrrhiza uralensis*
Neoliquiritin	51666248	*Glycyrrhiza uralensis*
protocatechuic acid	72	*Pseudocydonia sinensis*
gallic acid	370	*Pseudocydonia sinensis*
vanillic acid	8468	*Pseudocydonia sinensis*
catechin	9064	*Pseudocydonia sinensis*
oleanolic acid	10494	*Pseudocydonia sinensis*
syringic acid	10742	*Pseudocydonia sinensis*
ursolic acid	64945	*Pseudocydonia sinensis*
betulinic acid	64971	*Pseudocydonia sinensis*
epicatechin	72276	*Pseudocydonia sinensis*
betulin	72326	*Pseudocydonia sinensis*
erythodiol	101761	*Pseudocydonia sinensis*
Procyanidin B2	122738	*Pseudocydonia sinensis*
pomolic acid	382831	*Pseudocydonia sinensis*
p-coumaric acid	637542	*Pseudocydonia sinensis*
chlorogenic acid	1794427	*Pseudocydonia sinensis*
acetyl ursolic acid	6475119	*Pseudocydonia sinensis*
procyanidin B1	11250133	*Pseudocydonia sinensis*
aconitine	2012	*Aconitum carmichaeli*
hypaconitine	441737	*Aconitum carmichaeli*
mesaconitine	441747	*Aconitum carmichaeli*
deoxyaconitine	21598997	*Aconitum carmichaeli*
benzoylmesaconine (BMA)	24832659	*Aconitum carmichaeli*
Glc (glucose)	5793	*Atractylodes japonica*
ATO-III (atractylenolideIII)	155948	*Atractylodes japonica*
Fru (fructose)	445557	*Atractylodes japonica*
atractylodin	5321047	*Atractylodes japonica*
atractylodinol	10012964	*Atractylodes japonica*

**Table 2 nutrients-18-00008-t002:** Detailed information on the 6 core target genes identified from the Mecasin–Alzheimer’s disease network.

Target	Degree Centrality	Betweenness Centrality	Closeness Centrality
AKT1	36	0.013430632	0.928571429
STAT3	36	0.013961168	0.928571429
IL6	39	0.020930957	1.000000000
TNF	35	0.014525668	0.906976744
EGFR	35	0.013914223	0.906976744
IL1B	36	0.016267201	0.928571429

## Data Availability

The original contributions presented in this study are included in the article. Further inquiries can be directed to the corresponding author.

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
