# Peer review of "Network Pharmacology-Based Characterization of Mecasin (KCHO-1) as a Multi-Target Modulator of Neuroinflammatory Pathways in Alzheimer’s Disease"

_nutrients, 2025, doi:10.3390/nu18010008_

Round 1
Reviewer 1 Report
Comments and Suggestions for Authors
The study has volume, not depth. GeneCards + PubChem co-occurrence = basically every gene associated with inflammation, cancer, metabolism, etc. That’s why you got 942 overlapping genes - because your filtering method is overestimated.
“49.24% overlap between Mecasin targets and AD genes.” This number is biologically irrelevant because both datasets are massive and non-specific. Almost any herbal mixture will give you a 40–60% overlap.
Binding affinity in kJ/mol is wrong units. Docking programs output kcal/mol, not kJ/mol.
Your 14 “core genes” include:
GAPDH, EGFR, AKT1, IL6, TP53…
These are universal hubs, not disease-specific mechanisms. Any PPI-based study will spit out the same ones.
The Introduction is bloated. It repeats the same things 3 times: dementia is common, AD is multifactorial, inflammation is important - Cut it.
Repetitions: "Multi-target,” “network-based,” “systems-level,” “neuroinflammation,” “oxidative stress” - repeated.
GAPDH is a housekeeping gene. If GAPDH appears in your “core AD mechanism.
Reviewer 2 Report
Comments and Suggestions for Authors
Comments to the authors
Major comments
-The studied herbal compount contains several nutraceutics (Polygala tenuifolia, Gastrodia elata, Salvia miltiorrhiza, Paeonia lactiflora, Glycyrrhiza uralensis, Pseudocydonia sinensis, Aconitum carmichaeli, and Atractylodes japonica). Thus, it is not possible to elucidate the exactly molecular involved mechanisms in your study. In other words, inrteractions are consequence of several active principles from plants but without specific dependent effects of their nutraceuticals.
-The present study evaluates if Mecasin confer therapeutic benefit in AD by integratiing network-based pharmacological framework through multi-target mechanisms by Mecasin. However, these mechanisms are consequence of interaction between several active principles from plants and the exactly mechanism/s can not be demonstrated in your study.
-Although mecasin may exert neuroprotective and anti-inflammatory effects, the study really does not offer insights into its potential role as a multi-target candidate for AD intervention.
Since the herbal components of Mecasin has several active principles from plants (Curcuma longa, Polygala tenuifolia, Gastrodia elata, Salvia miltiorrhiza, Paeonia lactiflora, Glycyrrhiza uralensis, Pseudocydonia sinensis, Aconitum carmichaeli, and Atractylodes japonica), the study indicates that many genes are affected by Curcuma longa, 818 for Polygala tenuifolia, 322 for Gastrodia elata, 1,050 for 103 Salvia miltiorrhiza, 2,555 for Paeonia lactiflora, 2,719 for Glycyrrhiza uralensis, 1,289 for 104 Pseudocydonia sinensis, 329 for Aconitum carmichaeli, and 358 for Atractylodes japonica. However, these changes can not be explained by each active principle from plants and these could be justified by interactions between plants from the herbal composition.
In order to investigate the relevance of Mecasin to AD pathophysiology, AD-associated genes were retrieved from the GeneCards database (https://www.genecards.org/) using the term “Alzheimer’s Disease.” A total of 8,886 AD-related genes were collected. Although the molecular convergence of genes and signalling pathways in AD must be considered, this study does not explain a real causal effect between these active principles from plants and its beneficial effects in AD and it could be interactive effects between active principles from plants.
To visualize the multilevel interaction structure of Mecasin within the AD context, a drug–herb–compound–target–pathway (D-H-C-T-P) network was constructed.
How is possible to create this signalling pathway?
-Data obtained from compound screening, target identification, AD-related gene retrieval, and PPI–core gene analysis were integrated and imported into Cytoscape (version 3.10.1). Shall you give details on this process.
Minor comments
-The relevance of dementia in the introduction should be removed since the topic is not AD or other related-dementia. Remove the information about Korea as an orphan herbal investigational product for amyotrophic lateral sclerosis (ALS) or ALS patients [8]. Your study is not about ALS.
-The authors textually indicate ¨Because TNF is a secreted ligand and not consistently represented in proteome-wide co-regulation datasets, TRAF2 was employed as TNFR1/2–TRAF2 signalling, with results interpreted at the receptor–adaptor level rather than ligand abundance¨.
It is possible to demonstrate this degree of kwolegde without real molecular determination and only docking studies?
-For structural validation, molecular docking simulations were performed between mecasin-derived bioactive compounds and proteins encoded by the core genes. Docking was carried out using CB-Dock (http://clab.labshare.cn/cb-dock/php/index.php), which enables blind-docking cavity prediction and ligand-binding evaluation. Although the authors also optimize receptor conformations for ligand-binding analysis, there were absence of real molecular determinations that can confirm these afirmations.
-The Figure 1. Workflow of the network pharmacology analysis of Mecasin against Alzheimer’s disease 179 (AD) really does not perform this connexion and the visibility and quality is low. Please, improve it
The figure-2 A is impossible to see its content
At least, the table-2 List of overlapping target genes shared between Mecasin and Alzheimer’s disease gene 195 sets should be included within a section of supplementary materials but not within the mean text of the manuscript.
-Additionally, improve the quality of figure Figure 6. Drug–Herb–Compound–Target–Pathway (D–H–C–T–P) network of Mecasin in 263 Alzheimer’s disease (AD).
The quality of Figure 7 is low and difficult to see (fig-7. Co-expression, co-regulation, and molecular docking analyses of Mecasin– Alzheimer’s 291 disease core targets. (A) Protein–protein interaction (PPI) network showing strong associations 292 among TNF, IL6, and AKT1, with edge confidence values above 0.85).
The discussion is obvious and does not really explain their findings.
The conclusion should be shortened and improved it.
Comments on the Quality of English Language
The English is fine and does not require any improvement.
Round 2
Reviewer 1 Report
Comments and Suggestions for Authors
The authors fixed some technical issues (docking units, GAPDH, number of “core” genes, added a limitations paragraph), but the fundamental problems (huge nonspecific gene lists, over-interpretation of the 49% overlap, buzzwordy framing, volume>>depth) are still there.
The author still pull 1,913 targets from PubChem “chemical–gene co-occurrence” and intersect these with 8,886 AD genes from GeneCards. This yields 942 overlapping genes, which is 49.24% of their Mecasin target list.
In the Results and Discussion they describe this as a “substantial intersection” and claim it “highlights a strong mechanistic alignment between Mecasin’s pharmacological profile and AD pathological pathways” and “pharmacogenomic convergence.” With such inflated, literature-derived gene sets almost any polyherbal mixture or pleiotropic drug will give a 40–60% overlap. The number is dominated by list size, not biology.
Ideally the author should: apply a GeneCards score threshold or cross-validate with AD GWAS / curated AD gene panels, or restrict the analysis to specific processes (e.g. neuroinflammation, synaptic function) instead of the entire 8,886-gene omnibus list. If If they can’t/won’t, at minimum they must tone down all “strong alignment / convergence” language.
Interpretation of docking remains over-optimistic. The authors now use kcal/mol and cite a generic “< −6.0 kcal/mol” rule, and all values are between −7.7 and −11.7 kcal/mol. However, these are single docking runs on CB-Dock2, with no comparison to known ligands, no consideration of protein flexibility, and no ADME/BBB context.
Tone down the docking claims to “structural compatibility” / “plausible binding poses,” and avoid any implication of real binding affinities or pharmacological potency. Write a sentence stating that docking scores alone do not establish binding or efficacy, especially for multi-component herbal mixtures.
The Discussion repeats large parts of the Results almost verbatim (again walking through AKT1/STAT3/EGFR, the same KEGG pathways, then neuroregeneration, then inflammation).
“multi-target,” “multi-component,” “multi-pathway,” “systems-level,” “network-based,” “neuroinflammation,” “oxidative stress” appear over and over. Remove at least half of these buzzwords. Use ordinary wording
Wrong number of core genes in Methods 2.6. “Co-expression analysis was conducted in STRING to quantify the co-expression strength among the three core genes.” But there are six core genes throughout the rest of the paper.
“The workflow of the entire study is shown in Figure 1. we first curated their bioactive constituents…” – sentence starts with a lowercase “we” and “their” is unclear.
For only 6 core genes they report 589 BP terms, 144 KEGG pathways. This is not “wrong”, but it looks like over-fitting and gives a false sense of granularity.
I would recommend another major revision focusing on:
- more cautious interpretation of the overlap and core hubs,
- substantial condensation of the Introduction and Discussion,
- clearer emphasis that this is an exploratory data-mining exercise that generates hypotheses rather than providing mechanistic proof.”
Reviewer 2 Report
Comments and Suggestions for Authors
Dear Authors
Your responses are satisfactory for this reviewer.
Before the final acceptation, please, incude as limitation at the end of the discussion the follows sentence
The Mecasin herbal composition contains numerous phytochemicals and therefore could generates a large number of predicted compound–target associations as a limitation of your study
Additionally, add the abstract this sentence Each active priinciple in the herbal formulation (Mecasin) contains numerous phytochemicals that generate a large number of
predicted compound–target associations.
Decision: minor revision
